# Methyltransferases in the Pathogenesis of Keratinocyte Cancers

**DOI:** 10.3390/cancers13143402

**Published:** 2021-07-07

**Authors:** Eun Kyung Ko, Brian C. Capell

**Affiliations:** 1Department of Dermatology, Perelman School of Medicine at the University of Pennsylvania, Philadelphia, PA 19104, USA; EunKyung.Ko@pennmedicine.upenn.edu; 2Department of Genetics, Perelman School of Medicine at the University of Pennsylvania, Philadelphia, PA 19104, USA; 3Penn Epigenetics Institute, Perelman School of Medicine at the University of Pennsylvania, Philadelphia, PA 19104, USA; 4Abramson Cancer Center, Perelman School of Medicine at the University of Pennsylvania, Philadelphia, PA 19104, USA

**Keywords:** epigenetics, skin cancer, methylation, keratinocytes, cutaneous squamous cell carcinoma, basal cell carcinoma

## Abstract

**Simple Summary:**

Despite being the most common of all cancers, the underlying mechanisms behind the origins and progression of skin cancers like keratinocyte cancers are still emerging. Epigenetic dysregulation is pervasive in these cancers and it has become clear that by uncovering the mechanisms behind these changes, new therapeutic approaches may emerge.

**Abstract:**

Recent evidence suggests that the disruption of gene expression by alterations in DNA, RNA, and histone methylation may be critical contributors to the pathogenesis of keratinocyte cancers (KCs), made up of basal cell carcinoma (BCC) and cutaneous squamous cell carcinoma (cSCC), which collectively outnumber all other human cancers combined. While it is clear that methylation modifiers are frequently dysregulated in KCs, the underlying molecular and mechanistic changes are only beginning to be understood. Intriguingly, it has recently emerged that there is extensive cross-talk amongst these distinct methylation processes. Here, we summarize and synthesize the latest findings in this space and highlight how these discoveries may uncover novel therapeutic approaches for these ubiquitous cancers.

## 1. Introduction

Epigenetic regulation plays a critical role in the regulation of transcriptional activation and inactivation, and the dysregulation of these processes can play pathogenic roles in the progression of various diseases [1], including skin cancer. The major epigenetic changes observed at a cellular level include DNA methylation, post-translational modification of histones, and RNA modifications [2]. Aberrant epigenetic organization can alter the expression of genes involved in fundamental cellular pathways ranging from those involved in cell proliferation and metabolism to cell migration, motility, and apoptosis.

The three most common forms of skin cancers can be classified into two major types based on the cell of origin: melanoma and keratinocyte cancers (KCs), which include cutaneous squamous cell carcinoma (cSCC) and basal cell carcinoma (BCC) [3]. Melanoma originates from melanocytes, whereas the KCs, BCC, and cSCC, originate from epidermal keratinocytes. Collectively, KCs outnumber all other human malignancies [4]. BCC originates from basal cells near the epidermal–dermal junction or hair follicle, grows slowly, and rarely migrates or metastasizes. It is the most common cancer. cSCC can arise de novo or display stepwise progression from a precursor actinic keratosis (AK) to cSCC in situ (cSCCis), invasive cSCC, and finally metastatic cSCC [5]. cSCCs exhibit a metastatic risk between 0.1 and 13.7% [6], which occurs more frequently in the elderly and immunocompromised. With increasing incidence every year, cSCC is the second most common of all cancers and accounts for the majority of deaths (up to 5000–8000 a year) among KCs [7,8].

## 2. The Clinical and Mutational Landscape of KCs

Chronic ultraviolet radiation (UVR) exposure, immunosuppression, human papillomavirus (HPV) infection, and chronic cutaneous ulceration have been reported as primary causes for KCs [9]. BCC is typically locally invasive, it can also metastasize on rare occasions, and therefore treatment is critical for favorable long-term health outcomes. Mutations in members of the Hedgehog pathway such as Patched 1 (*PTCH1*) and Smoothened (*SMO*), as well as the cell cycle regulator, *TP53*, have been known to be the main drivers of BCC. One recent exome sequencing study sequenced 191 BCCs and 115 corresponding normal skin tissues. The sequencing revealed that *PTCH1* (58.6%), *TP53* (31.4%), and the *TERT* (59.2%) and *DPH3* (38.2%) promoters were highly mutated in BCCs [10]. Beyond these alterations, data from both The Cancer Genome Atlas (TCGA) [11,12] and Catalogue of Somatic Mutations in Cancer (COSMIC) [13] indicate that epigenetic regulators, and in particular histone methyltransferases, are also among the most commonly mutated genes. These include *KMT2D* (*MLL4*) (34–51%), *KMT2C* (*MLL3*) (40–47%), *KMT2A* (*MLL1*) (14–28%), *SETD1A* (10–21%), *NSD1* (23–24%), and *SETD2* (9–17%). The significant presence of these mutations underscores the critical role that epigenetic dysfunction may play in BCC pathogenesis.

Even more than BCC, cSCC is characterized as one of the most highly mutated forms of cancer in all of cancer biology. Long-term UVR exposure induces numerous CC → TT and C → T transitions, and the step-wise progression from AK to cSCC is well-established clinically. Similar to BCC, *TP53* mutations are pervasive in human cSCC, and in mice, the deletion of *Tp53* leads to increased rates of cSCC [14,15]. Along with p53 inactivation, *NOTCH1* mutations have been identified as another key driver in the early stages of cSCC carcinogenesis. Indeed, NOTCH1 is believed to act as a “gatekeeper” that blocks the transition to cSCC from normal skin [16]. Beyond these well-studied contributors to cSCC pathogenesis, like BCC, methyltransferases again display high rates of mutations. For example, *KMT2D* (*MLL4*) (42–52%), *KMT2C* (*MLL3*) (32–37%), *KMT2A* (*MLL1*) (19–30%), *SETD1A* (10–20%), *NSD1* (17–19%), *DNMT1* (12–17%), and *SETD2* (11–15%) are among the most highly mutated genes in cSCC. Notably, studies have identified that mutations in chromatin modifying enzymes such as these have been associated with both early clone formation in normal skin [17] as well as with aggressive and metastatic cSCC [18], highlighting the potentially critical role that normal methyltransferase function can play during both the initiation and progression of cSCC [19]. These findings emphasize both the importance of understanding the underlying mechanisms behind these observations, as well as the great potential that therapeutic targeting of the epigenome may offer for these incredibly common cancers.

## 3. DNA Methylation and Links to KCs

Aberrant DNA methylation associated with abnormal gene expression is one of the hallmarks of cancer. Both the silencing of tumor suppressor genes by hypermethylation and the activation of prometastatic genes or oncogenes by hypomethylation are characteristics of cancer cells16. DNA methyltransferases catalyze the addition of a methyl group (CH_3_) from its donor S-adenosine-methionine (SAM) to the cytosine nucleotide of DNA [20]. Three human DNA methyltransferases (DNMTs) have been identified: DNMT1, DNMT3A, and DNMT3B [21]. DNMT1 has been known to play a role in DNA methylation maintenance, while DNMT3A and 3B have roles in de novo DNA methylation [22,23,24].

In the skin, DNMT1 is expressed in the basal cells of the epidermis and is known to decrease with differentiation. As a result, it was found that DNA methylation was decreased in differentiated keratinocytes compared to epidermal basal cells. Consistent with this, the depletion of DNMT1 in human epidermal stem cells (EpSCs) promotes premature, irreversible differentiation [25]. Interestingly, another group reported that epidermis specific *Dnmt1* conditional knockout mice had uneven epidermal thickness and shorter hair follicle length than wild type mice [26]. The mice exhibited progressive alopecia with aging, suggestive of potential hair follicle stem cell exhaustion or reduced activation with aging.

Regarding DNMT3A and DNMT3B, one group generated mice with epidermis-specific deletions of both *Dnmt3a* and *Dnmt3b*, respectively. Intriguingly, they found that both Dnmt3a and Dnmt3b play important roles in regulating human EpSC homeostasis via their ability to promote enhancer activity of genes involved in self renewal and differentiation [27]. They then exposed the mice to the chemical carcinogen, DMBA/TPA, and found that mice lacking Dnmt3a, but not Dnmt3b, displayed epidermal squamous malignancies earlier than wild type mice, suggesting that Dnmt3a plays a tumor suppressive role in the epidermis. Notably, the combined deletion of both *Dnmt3a* and *Dnmt3b* led to more aggressive and metastatic cancers [28] (Figure 1).

Beyond the methyltransferases, the reversibility of DNA methylation has been demonstrated, and demethylases are considered promising targets for the development of therapeutics [29,30]. These include both the ten-eleven translocation (TET) methylcytosine dioxygenases [31], activation-induced cytidine deaminase (AID) [32], and thymine DNA glycosylase (TDG) [30,33,34]. It is likely that future studies will further elucidate how these demethylases may impact KC carcinogenesis given their demonstrated importance in other cancers and contexts.

Several global methylation profiles with human cSCC patients have been reported. One group analyzed the global DNA methylation status from AKs, cSCCs, and normal healthy epidermis and reported that the AK methylation pattern was similar to the cSCC methylation profile when compared to the normal epidermis. In addition, they demonstrated that there were two distinct sub-classes amongst the samples, one that displayed a more primitive methylome more consistent with embryonic stem cells while the other was more similar to healthy epidermis, suggesting two potentially unique origins [35]. Another study tested global DNA methylation profiles of cSCC biopsies taken from four different stages of cSCC (AK, early invasive carcinoma, high-risk non-metastatic carcinoma, and high-risk carcinoma with nodal metastasis). They reported that the initial invasive group displayed lower methylation levels than AK; however, high risk groups showed higher median methylation than the initial invasive group, indicating a gain of DNA methylation may be associated with the risk stage of Cscc [36] (Figure 1).

In addition to genome-wide methylation profiles, gene-specific DNA methylation changes have been correlated with gene expression alterations in cSCC [37]. Currently, hypermethylation resulting in transcriptional silencing at the promoters of numerous genes involved in cell cycle regulation, DNA repair, epithelial adhesion, and signal transduction has been shown to be a recurring pattern of DNA methylation changes related to cSCC [38]. Genes including cell cycle regulators such as *CDKN2A* [39], *CDH1* [40,41], T-cadherin (*CDH13*) [42], Forkhead Box E1 (*FOXE1*) [43], secreted frizzled-related proteins (*SFRP*s) [38], apoptosis-associated speck-like protein containing a caspase recruitment domain (*ASC*) [44], and G0/G1 switch gene 2 (*G0S2*) [45] are frequently methylated at the promoter regions in cSCC. Death associated protein kinase 1 (*DAPK1*) and Cadherin 13 (*CDH13*) are more highly methylated in invasive cSCC compared to both sun-exposed controls and sun-protected controls [46] (Figure 1). Recently it was reported that UVR exposure leads to upregulated DNMT1 and downregulated TETs in normal human and mouse skin tissues, and these changes induced hypermethylation of the inhibitor of DNA binding/differentiation 4 (*ID4*) gene involved in the TGFβ/BMP-SMAD-ID4 signaling pathway. The epigenetic silencing of *ID4* can lead to uncontrolled growth of cancer cells via this pathway, suggesting a role of ID4 silencing in tumorigenesis. Similarly, DNMT1 upregulation and TET downregulation accompanied *ID4* methylation in cSCC tissues, collectively suggesting that ID4 is downregulated by UVR irradiation via DNA methylation and acts as tumor suppressor gene in cSCC development [37].

The WNT signaling network has also emerged as an important regulator in the development and progression of KCs [47]. Secreted frizzled-related proteins (SFRPs) are known inhibitors of the WNT signaling pathway, binding WNTs directly to prevent receptor binding and the activation of WNT signaling [37]. The promoter of *SFRP3* was hypermethylated in human cSCC, suggesting that loss of SFRP3 and subsequent WNT activation led to metastatic cSCC [48]. In addition, other SFRP genes (*SFRP1*, *2*, *4*, and *5*) were likewise hypermethylated in cSCC compared to normal skin tissues [38,47]. In another study, the methylation status of 1505 CpG sites in primary and metastatic cSCC were compared. A CpG site in the promoter region of frizzled related protein (*FRZB*), a modulator of WNT signaling known to be involved in regulation of bone development, was hypermethylated in metastatic cSCC compared to non-metastatic primary cSCC (median methylation: 46.7% vs. 4.7%) [48]. The expression of WNT5A and WNT receptor frizzled homolog 6 (FZD6) have also been noted to be increased in cSCC [49]. These results, along with methylation pattern, highlight the potentially critical role of WNT signaling in the metastatic progression of cSCC (Figure 1).

In addition to hypermethylation, patterns of hypomethylation and concomitant gene overexpression have also been reported in cSCC. For example, one study demonstrated that DNA methylation levels of repetitive long interspersed nuclear elements-1 (LINE-1) were significantly lower in cSCCs compared to normal epidermal tissues using bisulfite restriction analysis [50] (Figure 1). Another study examined the expression level and methylation status of deletion in split hand/split foot 1 (*DSS1*) gene in 75 cSCC patients given the role of DSS1 overexpression in driving the early transformation of preneoplastic keratinocytes during chemical carcinogen-induced skin cancer [51]. Consistent with this, the authors found DSS1 to be both overexpressed and hypomethylated in 61 of the 75 patients (81.3%), and 12 out of remaining 14 had lower expression levels of DSS1 with heterogeneous methylation compared to normal skin tissues. The overexpression of DSS1 by hypomethylation was associated with poor cumulative overall survival rate and cumulative disease-free survival rate in cSCC [52]. Global hypomethylation with aging and sun exposure has been reported, further suggesting that these changes may occur before malignant transformation. However, the direct relationship between sun exposure-associated DNA hypomethylation and cSCC tumorigenesis remains to be confirmed.

While the differing models, study designs, and results described above make it hard to make broad conclusions about the role of DNA methylation in the cSCC pathogenesis, in general, a review of reported results suggests a model whereby hypomethylation may play an important role in the initial stages and transition from normal skin to cSCC. However, once cSCC is established, hypermethylation may in fact be a critical driver of progression and metastasis. We expect that additional future studies will continue to elucidate these underlying mechanisms.

With regards to BCCs, most studies have focused on a single or small number of genes. One group tested the DNA methylation status at the promoter regions of 9 tumor suppressor genes and one oncogene from 112 BCC patients and 124 healthy controls using methylation-specific PCR (MSP) [53]. Sonic hedgehog (*SHH*), adenomatous polyposis coli (*APC*), secreted frizzled-related protein 5 (*SFRP5*), and Ras association domain family 1 (*RASSF1*) genes were significantly hypermethylated in BCC compared to normal skin. Furthermore, they also reported increased WNT activity due to high levels of E-cadherin on the membrane in the BCC [53]. These data strongly suggest that epigenetic silencing through DNA hypermethylation may contribute to BCC tumorigenesis and could be a new target for treatment. In similar work looking at 52 BCCs, hypermethylation was identified at the promoter of *DCR2* (44%), *APC* (33%), *DCR1* (32.5%), *RASSF1* (32%), and *DAPK* (14%), respectively, in comparison to normal tissues. Together, these findings imply that altered DNA methylation may constitute an important pathway in the tumorigenesis of BCC [54]. However, the relatively small number of studies indicate the need for further investigation.

## 4. Histone Methylation: Highly Mutated Modifiers in KCs

Histones are divided into five types: H1, H2A, H2B, H3, and H4. A nucleosome is composed of a histone octamer core, which consists of two copies of each core histone H2A, H2B, H3, and H4 protein, and a short segment of DNA between 145 and 147 base pairs that is wrapped around it [55,56]. Theses nucleosome cores are stabilized by the linker histone H1, and each core has histone tails that interact with DNA [57,58]. Along histone tails, the lysine residue (K) is the main site of modifications, particularly acetylation and methylation. Histone H3 at lysine 4 (H3K4), lysine 9 (H3K9), lysine 27 (H3K27), lysine 36 (H3K36), and lysine 79 (H3K79), and histone H4 lysine 20 (H4K20) are well-established lysine methylation sites and can be mono-, di-, or trimethylated. Di- and trimethylation at H3K4, H3K36, and H3K79 are typically associated with gene activation, whereas H3K9, H3K27 and H4K20 trimethylation are generally associated with gene repression [59,60]. Twenty-four different lysine methyltransferases (KMTs) for the 6 lysine methylation sites have been reported to date: SETD1A, SETD1B, MLL1 (KMT2A), MLL2 (KMT2B), MLL3 (KMT2C), MLL4 (KMT2D) [61,62], SETD7, PRDM9 [63] for H3K4 methylation, SUV39H1, SUV39H2, SETDB1, G9A/EHMT2, GLP/EHMT1 for H3K39 methylation [64], SETD2, NSD1, NSD2, NSD3, ASH1L for H3K36 methylation [65], DOT1L for H3K79 methylation [66], SETD8, SUV420H1 and SUV420H2 for H4K20 methylation [67], and EZH1 and EZH2 for H3K27 methylation [68].

Mutations and copy number variations (CNVs) in histone methyltransferases have been shown to be frequent events in cancers, and particularly in cSCC [18,19,69,70]. These studies have suggested that mutations in chromatin modifying enzymes may also be particularly associated with poor overall survival in aggressive cSCC [19]. Consistent with this, another study performed exome and targeted sequencing with primary cSCC, metastatic cSCC, and normal skin tissues. The study revealed high mutations of epigenetic regulators: *KMT2D* (67%), *KMT2C* (58%), *SETD2* (50%), *KMT2A* (33%), and *KAT6A* (33%), in metastatic cSCC [18]. Interestingly however, only *KMT2D* (along with *TP53*) demonstrated significantly higher rates of mutation in metastatic tumors as compared to nonmetastatic primary tumors [18]. Intriguingly, another recent study looking at early preneoplastic clone formation in human skin showed that *KMT2D* mutations were also associated with these clones [17], collectively suggesting that *KMT2D* alterations may play critical roles in both the onset and progression of cSCC. Underscoring a potentially essential tumor suppressor role for KMT2D, recent work has shown that KMT2D is critical for the promotion of epidermal differentiation through its interaction with the master epithelial transcription factor, p63 [71], as well as through its ability to promote tumor suppressive ferroptosis, a form of iron-dependent programmed cell death [72,73] (Figure 2). In support of KMT2D’s role as a tumor suppressor in cSCC, inhibition of the histone demethylase enzyme that opposes KMT2D’s function, LSD1 (KDM1A), has been shown to both promote epidermal differentiation and suppress cSCC in 3D human cSCC organoids [74] (Figure 2). These results suggested that LSD1 inhibitors may be an effective pro-differentiation therapy for cSCC through their ability to promote increases in H3K4 methylation.

Enhancer of zest homolog 2 (EZH2), a histone methyltransferase for H3K27 trimethylation (H3K27me3), serves as a catalytic subunit of Polycomb repressive complex 2 (PRC2) that results in gene silencing via chromatin condensation [75]. The overexpression of EZH2 has been observed in a number of human cancers [76,77,78,79]. Clinical studies have reported that EZH2 expression levels are increased in dysplasia and malignant transformation and are associated with poor prognosis in both oral SCC and esophageal SCC [80,81,82,83,84], in part through its ability to its ability to suppress the expression of tumor suppressors such as p16 and p15, resulting in the upregulation of the cell cycle regulator, CCND1. This result is consistent with previous reports that the pharmacological inhibition of EZH2 leads to down-regulation of CCND1 in skin cancer, and CCND1 is linked to the malignant progression of oral premalignancy [85]. Finally, one study in cSCC cells suggested that EZH2 may repress innate immunity to repress antitumor immune responses and promote an increased risk of metastasis in cSCC [86].

*SETD2*, frequently mutated in KCs, is the only known gene in human cells responsible for the trimethylation of H3K36 (H3K36me3) and has been shown to be involved in transcriptional initiation and elongation, alternative splicing, and DNA damage repair [87,88]. Beyond these canonical SETD2 functions, SETD2 has recently been shown to also methylate non-histone targets such as cytoskeletal proteins including α-tubulin and F-actin [89,90,91,92]. In this role, a deficiency of SETD2 can provoke both mitotic defects and impaired cellular migration. In addition, SETD2 can bind p53, and in turn regulate the expression of select p53 target genes by enhancing the transcriptional activity of p53 [93]. Given that DNA-damaging UVR is the major carcinogen driving KCs development, dysfunction of SETD2 has been implicated in the development of both BCC and cSCC, though the underlying mechanisms are currently unknown. Given all the diverse ways in which SETD2 can impact cellular functions, it is likely that future studies will uncover ways in which these SETD2 mutations may be affecting the development of KCs.

Finally, one recent study reported the association between advanced BCC and the histone methyltransferase EZH2 [94,95]. First, the authors tested the expression levels of EZH2 and the proliferation marker Ki67 in 30 cases of less aggressive BCC histologic subtypes and 30 cases of more aggressive histologic subtypes (morpheaform, infiltrative, and micronodular). EZH2 and Ki67 expression were significantly increased in the more aggressive BCCs as compared to that of the less aggressive BCCs, and positively correlated with the aggressiveness of BCC. They suggested that EZH2 may be a potential target to inhibit BCC progression [95]. In follow-up work, they tested the expression levels of H3K27me3, 5hmC, NSD2, MOF, JARID1B, EZH2, and Ki67 from the same BCC patients that they examined in the previous study, as well as matched non-malignant normal epidermal tissue to further explore the relationship of epigenetic modifiers to BCC. They found that H3K27me3, 5hmC, NSD2, and MOF were upregulated and JARID1B was downregulated in BCCs as compared to non-malignant epidermal cells. Interestingly, H3K27me3 and 5hmC upregulation were positively correlated with a less aggressive subtype of BCC and negatively correlated with EZH2 in an aggressive BCC [95]. In contrast, EZH2 positively correlated with JARID1B expression in more aggressive BCC. Taken together, these results suggest that EZH2-associated epigenetic marker profiles can be utilized as histologic signs of BCC aggressiveness.

Collectively, these findings suggest a model whereby histone methyltransferases that promote gene expression undergo frequent loss of function and/or expression in cancer (i.e., KMT2D), while those that typically repress gene expression often are overexpressed or characterized by a gain of function (i.e., EZH2). While a great deal remains to be learned regarding the precise underlying mechanisms in KC pathogenesis, current evidence from a variety of cancers suggests that histone methyltransferase function may impact an array of different processes ranging from differentiation and DNA repair to metabolism and programmed cell death.

## 5. RNA Methylation: What’s Old Is New

Modifications upon RNA have been known of since the 1950s, and *N*^6^-methyladenosine (m^6^A), was both the first discovered, and the most abundant modification upon eukaryotic mRNAs [96]. Currently, there are over 170 known RNA modifications. However, it has only been within the last decade that the field has really exploded following the discovery that the m^6^A modification was reversible, thus setting the stage for the emergence of the “RNA epigenetics” or “epitranscriptomics” field [96]. It has been demonstrated that m^6^A is methylated by the heterodimer of methyltransferase-like 3 (METTL3) and methyltransferase-like 14 (METTL14), in complex with WTAP, KIAA1429, and ZFP217, and is demethylated by two demethylases, fat mass and obesity-associated protein (FTO) and ALKBH5 [97]. Additionally, methyltransferase-like 4 (METTL4) and Methyltransferase-like 16 (METTL16) have also shown to methylate the U2 small nuclear RNA (U2 snRNA) and U6 small nuclear RNA (U6 snRNA), respectively [97]. The m^6^A methylation on mRNA is thought to promote either mRNA degradation, or alternatively to enhance its stability and therefore its translation. These divergent outcomes are dependent upon both the “reader” proteins that bind to the m^6^A-modified mRNAs as well the cell type and cellular context.

Beyond these direct effects on mRNA, emerging data suggests increasing potential links between RNA modifications and other epigenetic gene regulatory mechanisms [98,99]. For example, a direct interaction between METTL14 and the histone modification, H3K36me3, was recently reported [100]. SETD2, which specifically catalyzes H3K36me3, showed a positive correlation with the mRNA expression levels of genes that make up the METTL3/14 complex. METT14 colocalized with H3K36me3 and RNA polymerase II in the nucleus, supporting the evidence that m^6^A deposition occurs co-transcriptionally. Additionally, other recent work has demonstrated how m^6^A is deposited on chromatin-associated RNAs (carRNAs), and that inhibiting this process leads to widespread increases in chromatin accessibility and aberrant transcription [98,101].

The dysregulation of the m^6^A modification has been observed in various human cancers, including skin cancers, although direct links have yet to be fully established between oncogenic or tumor suppressive functions of m^6^A and tumorigenesis. Recently, it was reported that METTL3 regulates the UVR induced DNA damage response. Specifically, DNA damage by UVR led to the rapid accumulation of m^6^A at damage sites. This METTL3-mediated m^6^A enhanced the survival of cells after UVR exposure, while its loss led to an increased sensitivity to UVR. They found unexpectedly that these observations were driven by the ability of the m^6^A modification to facilitate the repair of the UVR damaged DNA and thus enhance cell survival [102]. Notably, these studies were limited to UVC radiation and cancer cell lines, and therefore future studies into the role of METTL3-mediated m^6^A in UVA and UVB responses in relevant skin models will further elucidate these potential connections to skin cancer pathology.

Several studies have shown that METTL3 can serve as an oncogene in various cancers such as gastric, bladder, colon, pancreatic and glioblastoma [100]. Recently, it was reported that the expression of METTL3 in cSCC was significantly upregulated and that it promoted the expression of the ΔNp63 isoform of p63, which enhanced cSCC cell proliferation and tumor growth [103] (Figure 3). This report is consistent with previous results showing that ΔNp63, which exerts oncogenic properties, was amplified in human cSCCs. Furthermore, depletion of METTL3 impaired cSCC cell stem-like properties such as colony forming ability in vitro and tumorigenicity in vivo [103]. These notable results support a link between ΔNp63 and METTL3 in cSCC tumorigenesis. In line with this, another study showed that some of the most downregulated genes with METTL3 depletion are the basal EpSC genes BPAG1 (*COL17A1*) and integrin b4 (*ITGB4*) [104]. Another group recently reported that an epidermal *Mettl14* conditional knockout mouse shows aberrant skin development and impaired wound healing, including a thicker epidermis and a reduced number of p63-positive basal cells. Specifically, Mettl14-dependent m^6^A on the lncRNA, plasmacytoma variant translocation 1 (*Pvt1*), a potential oncogenic lncRNA associated with MYC, was revealed to support epidermal stemness by enhancing its interaction with MYC [105] (Figure 3). At this time, the role of RNA modifications in BCC pathogenesis has not been explored, though given recent evidence implicating METTL3’s role in hair follicle development and WNT signaling [104] (Figure 3), one can surmise that it will likely impact BCC biology as well.

While there have been a limited number of studies of RNA methylation in KCs, the related cancer of head and neck SCC (HNSCC) has been the subject of a number of investigations in the field. For example, one study demonstrated that METTL3 increased the stability of cMyc via the reader, YTH *N*6-methyladenosine RNA binding protein 1 (YTH) domain family member 1 (YTHDF1) to enhance tumorigenesis [106]. Consistent with this, METTL3 was found to be increased in expression in comparison to adjacent normal mucosa tissues and was associated with poor prognosis by enhancing proliferation capacity, migration, and invasion in oral HNSCC. Similarly, another study provided evidence to suggest that METTL3 overexpression cooperates with the reader IGF2BP1 to promote the mRNA translation of *BMI1* to promote HNSCC carcinogenesis [107]. The authors showed that tumor xenografts with METTL3 knockdown cells showed significantly smaller tumor sizes and volumes compared with non-targeted shRNA control cells in nude mice. Likewise, utilizing the 4NQO (4-nitroquinoline-1-oxide) chemical carcinogen model of HNSCC, they found that *Mettl3* wild-type mice had 100% penetrance of dysplasia and SCC, while only a third of *Mettl3* knockout mice developed mild dysplasia [107]. Collectively, these in vitro and in vivo results strongly support that METTL3 may act as an oncogene and promote tumorigenesis in both cSCC and HNSCC and highlight the potential of targeting aberrant RNA methylation for cancer therapy.

## 6. Conclusions and Future Perspectives

The methylation of DNA, histones, and RNA are dynamic epigenetic modifications that play central roles in the balance between cancer prevention and progression. The emerging studies, particularly in the area of DNA and histone methylation, have identified clear associations between dysregulated epigenetics and various cancers, including KCs. This has identified both potential biomarkers of disease as well as potential therapeutic targets. Notably, with the recent emergence of reversible RNA methylation and epitranscriptomics, it is likely that new potential targets will continue to emerge. The potentially significant roles of RNA modifications in a variety of cancers, including multiple forms of SCC, suggests that future studies will reveal mechanisms by which they too contribute to KC pathogenesis. Together, these findings have provided a better understanding of the interplay between DNA, histone, and RNA methylation in carcinogenesis and the role they may play in promoting KCs. Despite this, future studies will undoubtedly need to address the significant gaps in our understanding and more directly test the potential safety, utility, and efficacy of targeting these pathways, including their ability to synergize with other more established anticancer agents.

## Figures and Tables

**Figure 1 cancers-13-03402-f001:**
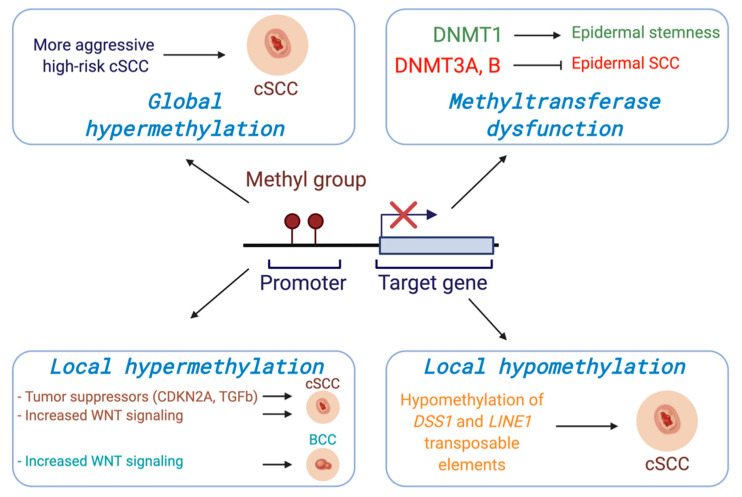
DNA methylation at gene promoters generally serves to suppress expression. Alterations in DNA methylation can occur both globally across the genome or more locally at specific loci in cancer and can drive tumorigenesis through transcriptional dysregulation. Consistent with a key role for DNA methylation in KCs, in vivo models have shown that DNA methyltransferases are critical regulators in the balance between epidermal differentiation and tumorigenesis.

**Figure 2 cancers-13-03402-f002:**
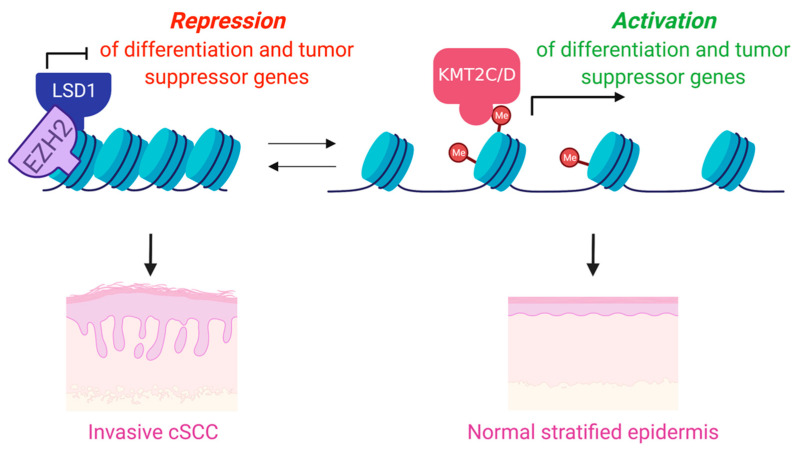
While the H3K4 methyltransferases have been shown to play roles in the promotion of differentiation and tumor suppression in the epidermis, the demethylases have been shown repress differentiation and are frequently overexpressed in KCs.

**Figure 3 cancers-13-03402-f003:**
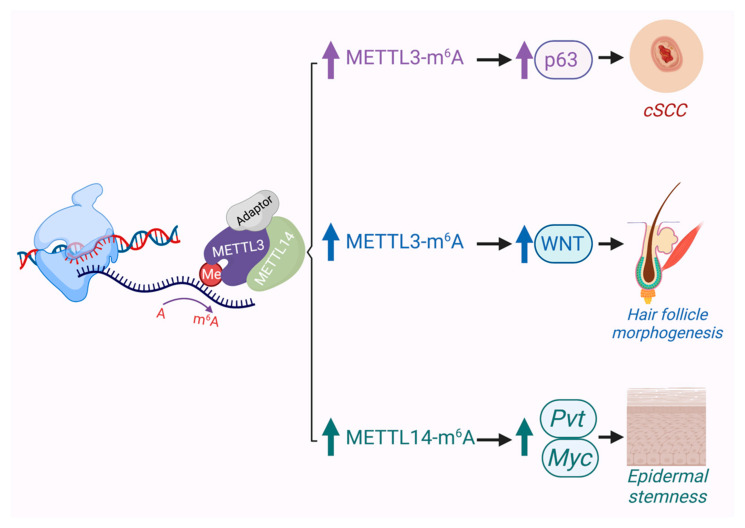
In the skin, evidence suggests that METTL3/14-mediated m6A can play critical roles in development of hair follicles, promoting the epidermal stem cell state, and driving cSCC when overexpressed.

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
