# Peer review of "Methyltransferases in the Pathogenesis of Keratinocyte Cancers"

_cancers, 2021, doi:10.3390/cancers13143402_

Round 1
Reviewer 1 Report
The manuscript “Methyltransferases in the pathogenesis of keratinocyte cancers” by Eun Kyung Ko et al. summary the roles of methyltransferases in diseases. The manuscript provides useful information for basic research and clinical application. Authors may mention the relationship between the defects of methyltransferases and keratinocyte cancers.
Author Response
We thank the reviewers for their thoughtful comments and reviews (below in black italic text). Below we provide a point-by-point response (blue text) and new updated aspected of the revised manuscript appear in red. We believe these reviews have substantially improved the manuscript.
Reviewer 1
The manuscript “Methyltransferases in the pathogenesis of keratinocyte cancers” by Eun Kyung Ko et al. summary the roles of methyltransferases in diseases. The manuscript provides useful information for basic research and clinical application. Authors may mention the relationship between the defects of methyltransferases and keratinocyte cancers.
We thank the reviewer for appreciating our summary of the major discoveries in the field. We have gone through the manuscript again and added all the published evidence, to our knowledge, that links methyltransferase dysregulation and dysfunction to keratinocyte cancers.
Reviewer 2 Report
The manuscript “Methyltransferases in the pathogenesis of keratinocyte cancers” by Eun Kyung Ko and Brian C. Capell, submitted to Cancers, summarizes current knowledge on the pathogenesis of keratinocyte cancers. This is a very interesting problem and the authors succeeded in creating a comprehensive review shading light on this question. In the submitted manuscript the authors summarized the current knowledge in this field, in particular, the authors focused on the role of hyper- and hypomethylation responsible for the silencing of tumor suppressor genes and activation of prometastatic genes respectively. The authors conclude that the balance between cancer prevention and progression can be controlled by the state of methylation of DNA, histones, and RNA.
The manuscript is well written and clearly illustrated. Undoubtedly, this review will be met with a vivid interest by the scientific society. The manuscript can be accepted in present form.
Minor comments
Line 84. Reference 16 is without brackets.
Author Response
We thank the reviewers for their thoughtful comments and reviews (below in black italic text). Below we provide a point-by-point response (blue text) and any new changes to the manuscript appear in red. We believe these reviews have substantially improved the manuscript.
Reviewer 2
The manuscript “Methyltransferases in the pathogenesis of keratinocyte cancers” by Eun Kyung Ko and Brian C. Capell, submitted to Cancers, summarizes current knowledge on the pathogenesis of keratinocyte cancers. This is a very interesting problem and the authors succeeded in creating a comprehensive review shading light on this question. In the submitted manuscript the authors summarized the current knowledge in this field, in particular, the authors focused on the role of hyper- and hypomethylation responsible for the silencing of tumor suppressor genes and activation of prometastatic genes respectively. The authors conclude that the balance between cancer prevention and progression can be controlled by the state of methylation of DNA, histones, and RNA.
The manuscript is well written and clearly illustrated. Undoubtedly, this review will be met with a vivid interest by the scientific society. The manuscript can be accepted in present form.
We very much appreciate the reviewer’s appreciation of the comprehensive nature of our review, as well as the writing and illustrations and how it will be received.
Reviewer 3 Report
I read with interest this Review manuscript regarding the role of methylation in keratinocyte skin cancers.
The aims are to review published work on DNA methylation, Histone methylation, and RNA methylation and their purported implications. This is largely influenced by work from the authors laboratory, and supported with external evidence.
In its current form, I would support support its publication with the following caveats.
Major criticisms
1. The review is written largely as a descriptive summary of published work, but does not attempt to synthesise diverse findings together. This is of course made difficult by the fact that individual studies on methylation are diverse, and can be conflicting in proposed mechanism and effect size.
2. Through the text, the proposed “new targets” for pharmaceutical modulation have not been validated in any clinical studies and their likelihood of uptake are overplayed. Although several pre-clinical studies have hinted at mechanisms and potential use, I believe the utility here has been overstated. I would recommend toning the utility aspect to reflect that this is ongoing. We are still unsure if methylation changes are causative or bystander effects.
Minor criticisms
1. Pg 1 Line 39-41 Not all cSCCs exhibit a clear progression from AK to metastatic cSCC.
2. Pg 2 Line 48 I would not describe BCCs as “just” locally invasive.
3. Figure 2 is largely the same as published Fig in Reference 77 by the same authors.
4. Figure 3 is not referenced in the text.
5. There are several referencing errors eg. No journal name in Ref 42, 45, or 46, and incorrect details in Ref 77. I have not exhaustively reviewed references.
Author Response
We thank the reviewers for their thoughtful comments and reviews (below in black italic text). Below we provide a point-by-point response (blue text), and any new edits to the manuscript appear in red. We believe these reviews have substantially improved the manuscript.
Reviewer 3
I read with interest this Review manuscript regarding the role of methylation in keratinocyte skin cancers.
The aims are to review published work on DNA methylation, Histone methylation, and RNA methylation and their purported implications. This is largely influenced by work from the authors laboratory, and supported with external evidence.
In its current form, I would support its publication with the following caveats.
Major criticisms
1. The review is written largely as a descriptive summary of published work, but does not attempt to synthesise diverse findings together. This is of course made difficult by the fact that individual studies on methylation are diverse, and can be conflicting in proposed mechanism and effect size.
We thank the reviewer for their careful reading and feedback on the manuscript. We appreciate this comment and have now gone through the review and added some short paragraphs to try to synthesize and highlight some potential underlying shared mechanisms and principles that may be inferred from the current existing data.
- Through the text, the proposed “new targets” for pharmaceutical modulation have not been validated in any clinical studies and their likelihood of uptake are overplayed. Although several pre-clinical studies have hinted at mechanisms and potential use, I believe the utility here has been overstated. I would recommend toning the utility aspect to reflect that this is ongoing. We are still unsure if methylation changes are causative or bystander effects.
This is a really excellent point. We have now taken steps to clarify and tone down our statements on therapeutic potential, in order to better reflect that field is still in a relatively immature state with regards to our understanding of how efficacious these therapeutic approaches may prove to be ultimately and there is extensive work that remains to be done.
Minor criticisms
1. Pg 1 Line 39-41 Not all cSCCs exhibit a clear progression from AK to metastatic cSCC.
Thank you for this clarification, we have corrected this in the text.
- Pg 2 Line 48 I would not describe BCCs as “just” locally invasive.
This is an important point and we have corrected the text to reflect that.
Figure 2 is largely the same as published Fig in Reference 77 by the same authors.
We thank the reviewer for noticing this unintentional similarity to another figure. However, we were not entirely clear on what figure or reference the reviewer was referring to, as reference 77 on the submitted form of the manuscript is a recent BioRxiv submission that to our knowledge does not have any figures similar to those in this review. We are happy to make any adjustments or changes once we find out exactly which figure the reviewer is referring to.
Figure 3 is not referenced in the text.
We thank the reviewer for noticing this and we have corrected this oversight.
There are several referencing errors eg. No journal name in Ref 42, 45, or 46, and incorrect details in Ref 77. I have not exhaustively reviewed references.
We thank the reviewer for noticing these errors and have gone through all the references to ensure they are all correct.